# Local Structure of Convex Surfaces near Regular and Conical Points

Alexander Plakhov [1,2]

1    Center for R & D in Mathematics and Applications, Department of Mathematics, University of Aveiro, 3810-193 Aveiro, Portugal; plakhov@ua.pt

2    Institute for Information Transmission Problems, 127051 Moscow, Russia

**Abstract:** Consider a point on a convex surface in $\mathbb{R}^d$, $d \geq 2$ and a plane of support $\Pi$ to the surface at this point. Draw a plane parallel with $\Pi$ cutting a part of the surface. We study the limiting behavior of this part of the surface when the plane approaches the point, being always parallel with $\Pi$. More precisely, we study the limiting behavior of the normalized surface area measure in $S^{d-1}$ induced by this part of the surface. In this paper, we consider two cases: (a) when the point is regular and (b) when it is singular conical, that is the tangent cone at the point does not contain straight lines. In Case (a), the limit is the atom located at the outward normal vector to $\Pi$, and in Case (b), the limit is equal to the measure induced by the part of the tangent cone cut off by a plane.

**Keywords:** convex surfaces; surface area measure of a convex body; Newton's problem of minimal resistance

## 1. Introduction

Consider a convex compact set $C$ with a nonempty interior in Euclidean space $\mathbb{R}^d$, $d \geq 2$. Let $r_0 \in \partial C$ be a point on its boundary, and let $\Pi$ be a plane of support to $C$ at $r_0$. Consider the part of the boundary $\partial C$ containing $r_0$ and bounded by a plane parallel with $\Pi$. We are interested in studying the limiting properties of this part of the boundary when the bounding plane approaches $\Pi$.

In what follows, a convex compact set with a nonempty interior will be called a convex body.

The point $r_0 \in \partial C$ is called *regular* if the plane of support at this point is unique and *singular* otherwise. It is well known that regular points form a full-measure set in $\partial C$.

Let $e$ denote the outward unit normal vector to $\Pi$. Take $t > 0$, and let $\Pi_t$ be the plane parallel with $\Pi$ at the distance $t$ from it, on the side opposite to the normal vector. Thus, the plane $\Pi = \Pi_0$ is given by the equation $\langle r - r_0, e \rangle = 0$ and $\Pi_t$ by the equation $\langle r - r_0, e \rangle = -t$. The body $C$ is contained in the closed half-space $\{r : \langle r - r_0, e \rangle \leq 0\}$. Here and in what follows, $\langle \cdot, \cdot \rangle$ means the scalar product.

Consider the convex body:

$$C_t = C \cap \{r : \langle r - r_0, e \rangle \geq -t\}.$$

In other words, $C_t$ is the part of $C$ cut off by the plane $\Pi_t$. The boundary of $C_t$ is the union of the convex set of codimension 1:

$$B_t = C \cap \{r : \langle r - r_0, e \rangle = -t\} \tag{1}$$

and the convex surface:

$$S_t = \partial C \cap \{r : \langle r - r_0, e \rangle \geq -t\}; \tag{2}$$

thus, $\partial C_t = B_t \cup S_t$.

In what follows, we will denote as $|\mathcal{A}|_m$ the $m$-dimensional Hausdorff measure of the Borel set $\mathcal{A} \subset \mathbb{R}^d$. By default, $|\cdot|$ means $|\cdot|_{d-1}$.

Let $n_r$ denote the outward unit normal to $C$ at a regular point $r \in \partial C$, and let $S$ be a Borel subset of $\partial C$. *The surface area measure induced by $S$* is the Borel measure $\nu_S$ defined in $S^{d-1}$ satisfying

$$\nu_S(\mathcal{A}) := |\{r \in S : n_r \in \mathcal{A}\}|$$

for any Borel subset $\mathcal{A} \subset S^{d-1}$. In the case when $S$ coincides with $\partial C$, we obtain the well-known measure $\nu_{\partial C}$ called the *surface area measure of the convex body $C$*. For this measure, the following well-known relation takes place:

$$\int_{S^{d-1}} n\, \nu_{\partial C}(dn) = \vec{0}. \tag{3}$$

Denote by $\nu_t$ the normalized measure induced by the surface $S_t$; more precisely,

$$\nu_t := \frac{1}{|B_t|} \nu_{S_t}.$$

That is, for any Borel set $\mathcal{A} \subset S^{d-1}$, it holds

$$\nu_t(\mathcal{A}) = \frac{1}{|B_t|} |\{r \in S_t : n_r \in \mathcal{A}\}|.$$

The surface area measure of $\partial C_t$ equals $\nu_{\partial C_t} = |B_t|\delta_{-e} + |B_t|\nu_t$; hence,

$$\int_{S^{d-1}} n\, d\nu_{\partial C_t}(n) = |B_t| \left(-e + \int_{S^{d-1}} n\, \nu_t(dn)\right).$$

Here and in what follows, $\delta_e$ means the unit atom supported at $e$. Applying Formula (3) to $\partial C_t$, one obtains

$$\int_{S^{d-1}} n\, \nu_t(dn) = e. \tag{4}$$

We say that $\nu_t$ *weakly converges* to $\nu_*$ as $t \to 0$ and denote $\lim_{t \to 0} \nu_t = \nu_*$, if for any continuous function $f$ on $S^{d-1}$, it holds

$$\lim_{t \to 0} \int_{S^{d-1}} f(n)\, \nu_t(dn) = \int_{S^{d-1}} f(n)\, \nu_*(dn).$$

Similarly, we say that $\nu_*$ is a *weak partial limit* of the measure $\nu_t$, if there exists a sequence of positive numbers $t_i$, $i \in \mathbb{N}$ converging to 0 such that, for any continuous function $f$ on $S^{d-1}$, it holds

$$\lim_{i \to \infty} \int_{S^{d-1}} f(n)\, \nu_{t_i}(dn) = \int_{S^{d-1}} f(n)\, \nu_*(dn).$$

In this article, we are going to study the limiting properties of the measure $\nu_t$ as $t \to 0$.

One such property is derived immediately. Let $\nu_*$ be a weak limit or a weak partial limit of $\nu_t$. Passing to the limit $t \to 0$ or to the limit $t_i \to 0$ in Formula (4), one obtains

$$\int_{S^{d-1}} n\, \nu_*(dn) = e. \tag{5}$$

The *tangent cone* to $C$ at $r_0 \in \partial C$ is the closure of the union of all rays with vertex at $r_0$ that intersect $C \setminus r_0$. Equivalently, the tangent cone at $r_0$ is the smallest closed cone with the vertex at $r_0$ that contains $C$; see Figure 1.

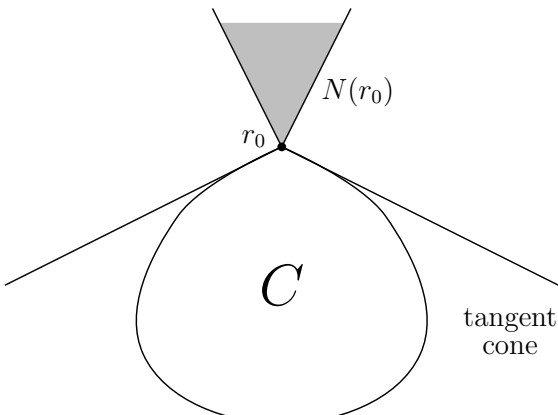

**Figure 1.** The tangent cone and the normal cone to a convex body $C$.

If the tangent cone at $r_0$ is a half-space, then the point $r_0$ is regular, and vice versa.

The *normal cone* to $C$ at $r_0$ is the union of all rays with vertex at $r_0$ whose director vector is the outward normal to a plane of support at $r_0$. It is denoted as $N(r_0)$. An equivalent definition is the following: the normal cone at $r_0$ is the set of points $r$ that satisfy $\langle r - r_0, r' - r_0 \rangle \leq 0$ for all $r' \in C$. The normal cone to a convex body does not contain straight lines. Both tangent and normal cones are, of course, convex sets.

If the dimension of $N(r_0)$ equals $d$ (equivalently, if the tangent cone does not contain straight lines), then $r_0$ is called a *conical point* of $C$. If the dimension of $N(r_0)$ equals 1, then $r_0$ is regular, and vice versa. In the intermediate case, that is if the dimension of $N(r_0)$ is greater than 1, but smaller than $d$, $r_0$ is called a *ridge point*. This notation goes back to Pogorelov [1].

The motivation for this study comes, to a great extent, from extremal problems in classes of convex bodies and, in particular, from Newton's problem of least resistance for convex bodies [2]. It is natural to try to develop a geometric method of the small variation of convex bodies for such problems, and perhaps, the simplest way would be cutting a small part of the body by a plane. This method proved itself to be effective in the case of Newton's problem. Let us describe this problem in some detail.

The problem in a class of radially symmetric bodies was first stated and solved by Newton himself in 1687 in [3]. The more general version of the problem was posed by Buttazzo and Kawohl in 1993 in [2]. This general problem can be formulated in the functional form as follows:

Find the smallest value of the functional

$$\iint_\Omega \frac{1}{1 + |\nabla u(x,y)|^2} \, dx dy \tag{6}$$

in the class of convex functions $u : \Omega \to \mathbb{R}$ satisfying $0 \leq u \leq M$, where $\Omega \subset \mathbb{R}^2$ is a planar convex body and $M > 0$.

The physical meaning of this problem is as follows: find the optimal streamlined shape of a convex body moving downwards through an extremely rarefied medium, provided that the body–particle collisions are perfectly elastic.

Problem (6) (along with its further generalizations) has been studied in various papers including [4–13], but has not been solved completely until now.

It was conjectured in 1995 in [6] that the slope of the graph of an optimal function near the zero level set $L_0 = \{(x,y) : u(x,y) = 0\}$ equals 1. This conjecture was numerically disproven by Wachsmuth (personal communication) in the case when $L_0$ has an empty interior and, therefore, is a line segment. Moreover, numerical simulation shows that the infimum of $|\nabla u|$ in the complement of $L_0$ is strictly greater than 1.

On the other hand, this conjecture was proven by the author in [14] in the case when $L_0$ has a nonempty interior. More precisely, it was proven that if $u$ minimizes functional (6), then for almost all $(x, y) \in \partial L_0$, it holds

$$\lim_{(x',y')\,(\notin L_0) \to (x,y)} |\nabla u(x', y')| = 1.$$

The proof is based on the results concerning local properties of convex surfaces near ridge points in the case $d = 3$. These results were formulated, with the proofs being briefly outlined, in [14].

**Remark 1.** *The limiting behavior of $\nu_t$ in the case $d = 2$ is quite simple. In this case, the tangent cone is an angle, which degenerates to a half-plane if the point is regular. We will call it the* tangent angle. *Let the tangent angle to $C \subset \mathbb{R}^2$ at $r_0$ be given by*

$$\langle r - r_0,\, e_1 \rangle \leq 0, \quad \langle r - r_0,\, e_2 \rangle \leq 0, \quad |e_1| = 1,\ |e_2| = 1,$$

*and e be given by*
$$e = \lambda_1 e_1 + \lambda_2 e_2, \quad \lambda_1 \geq 0,\ \lambda_2 \geq 0, \quad |e| = 1.$$

*Thus, $e_1$ and $e_2$ are the outward unit normals to the sides of the angle, and e is the outward unit normal to a line of support at $r_0$. Then, the limiting measure is the sum of two atoms:*

$$\lim_{t \to 0} \nu_t = \lambda_1 \delta_{e_1} + \lambda_2 \delta_{e_2}.$$

*The proof of this relation is simple and is left to the reader.*

*Note that if the point $r_0$ is regular, then $e_1 = e_2 = e$. It may also happen that the point is singular, that is $e_1 \neq e_2$, and e coincides with one of the vectors $e_1$ and $e_2$. In both cases, the limiting measure is an atom:*

$$\lim_{t \to 0} \nu_t = \delta_e.$$

The limiting behavior of $\nu_t$ is different for different kinds of points:

(a) If the point $r_0$ is regular, then the limiting measure is an atom.

(b) If $r$ is a conical point, then the limiting measure coincides with the measure induced by the part of the boundary of the tangent cone cut off by a plane $\Pi_t$, $t = \sigma$ (note that all the induced measures with $t > 0$ are proportional).

(c) The case of ridge points is the most interesting. In this case, the limiting measure may not exist, and the characterization of all possible partial limits is a difficult task.

Still, the study is nontrivial also in Cases (a) and (b). In this paper, we restrict ourselves to these cases, while Case (c) is postponed to the future. The main results of the paper are contained in the following Theorems 1 and 2.

**Theorem 1.** *If $r_0$ is a regular point of $\partial C$, then*

$$\lim_{t \to 0} \nu_t = \delta_e. \tag{7}$$

Let $r_0$ be a conical point, $K$ be the tangent cone at $r_0$, $\hat{S}_t$ be the part of $\partial K$ containing $r_0$ cut off by the plane $\Pi_t$, and $\hat{B}_t$ be the intersection of the cone with the cutting plane $\Pi_t$, $t > 0$, that is

$$\hat{S}_t = \partial K \cap \{r : \langle r - r_0,\, e \rangle \geq -t\} \quad \text{and} \quad \hat{B}_t = K \cap \{r : \langle r - r_0,\, e \rangle = -t\}.$$

Let $K_t = K \cap \{r : \langle r - r_0,\, e \rangle \geq -t\}$ be the part of the cone cut off by the plane $\Pi_t$; its boundary is $\partial K_t = \hat{S}_t \cup \hat{B}_t$.

All measures induced by $\hat{S}_t$ are proportional, that is the measure:

$$\nu_\star := \frac{1}{|\hat{B}_t|} \nu_{\hat{S}_t}$$

does not depend on $t$.

**Theorem 2.** *If $r_0$ is a conical point of $\partial C$, then*

$$\lim_{t \to 0} \nu_t = \nu_\star. \tag{8}$$

## 2. Proof of Theorem 1

The proof is based on several propositions.

Consider a convex set $D \subset \mathbb{R}^{d-1}$, and let $A = |D|$ be its $(d-1)$-dimensional volume and $P = |\partial D|_{d-2}$ be the $(d-2)$-dimensional volume of its boundary.

**Proposition 1.** *If $D$ contains a circle of radius a, then*

$$P \le \frac{d-1}{a} A. \tag{9}$$

**Proof.** Let $ds \subset \partial D$ be an infinitesimal element of the boundary of $D$ and denote by $p(ds)$ its $(d-2)$-dimensional volume. Consider the pyramid with the vertex at the center $O$ of the circle and with the base $ds$, that is the union of line segments joining $O$ with the points of $ds$. Let $A(ds)$ be the element of the $(d-1)$-dimensional volume of this pyramid; see Figure 2. Then, we have

$$A(ds) \ge \frac{a}{d-1} p(ds),$$

and therefore,

$$A = \int_{\partial D} A(ds) \ge \frac{a}{d-1} \int_{\partial D} p(ds) = \frac{a}{d-1} P.$$

From here follows Inequality (9). $\square$

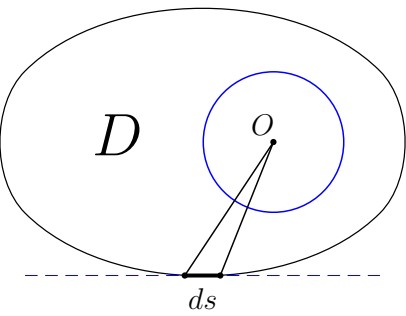

**Figure 2.** The convex set $D$ containing a circle of radius $a$.

Consider Euclidean space $\mathbb{R}^d$ with the coordinates $(x, z)$, $x = (x_1, \ldots, x_{d-1})$, and fix $t > 0$ and $0 < \varphi < \pi/2$.

**Proposition 2.** *Let a convex body $C \subset \mathbb{R}^d$ be contained between the planes $z = 0$ and $z = t$, $t > 0$. Let $D$ be the image of $C$ under the natural projection of $\mathbb{R}^d$ on the x-plane, $(x, z) \mapsto x$, and let $P = |\partial D|_{d-2}$ be the $(d-2)$-dimensional volume of $\partial D$. Let a domain $\mathcal{U} \subset \partial C$ be such that the outward normal $n_r = (n_{r,1}, \ldots n_{r,d-1}, n_{r,d})$ at each regular point $r \in \mathcal{U}$ satisfies $|n_{r,d}| \le \cos \varphi$. (In other words, the angles between $n_r$ for $r \in \mathcal{U}$ and the vectors $\pm(0, \ldots, 0, 1)$ are $\ge \varphi$.) Then,*

$$|\mathcal{U}| \le \frac{2tP}{\sin \varphi}.$$

**Proof.** The body $C$ is bounded below by the graph of a convex function, say $u_1$, and above by the graph of a concave function, say $u_2$; see Figure 3. Both functions are defined on $D$. That is, we have

$$C = \{(x, z) : x \in D, \ u_1(x) \leq z \leq u_2(x)\}.$$

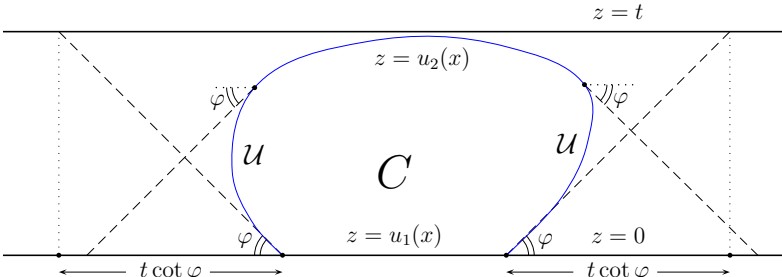

**Figure 3.** The body $C$ between two parallel planes $z = 0$ and $z = t$ is shown. Here, $\mathcal{U}$ is represented by the union of two curves bounded by the points.

Let $\mathcal{U}_i$ ($i = 1, 2$) denote the intersection of $\mathcal{U}$ with the graph of $u_i$. Clearly, if a point $(x, u_i(x))$ is regular and belongs to $\mathcal{U}_i$, then $|\nabla u_i(x)| \geq \tan \varphi$.

For $0 \leq z \leq t$, denote by $P_z$ the $(d - 2)$-dimensional volume of the set:

$$L_z = \{x : u_1(x) = z \text{ and } (x, u_1(x)) \in \mathcal{U}_1\}.$$

One clearly has $P_z \leq P$. Let $s$ be the $(d - 2)$-dimensional parameter in $L_z$, and let $ds$ be the element of the $(d - 2)$-dimensional volume in $L_z$. Denote by $x(z, s)$ the point in $L_z$ corresponding to the parameter $s$. Then, the $(d - 1)$-dimensional volume of $\mathcal{U}_1$ equals

$$|\mathcal{U}_1| = \int_0^t dz \int_{L_z} \sqrt{1 + \frac{1}{|\nabla u_1(x(z,s))|^2}} \, ds \leq \int_0^t P_z \sqrt{1 + \cot^2 \varphi} \, dz \leq \frac{tP}{\sin \varphi}.$$

The same argument holds for $\mathcal{U}_2$. It follows that $|\mathcal{U}| = |\mathcal{U}_1| + |\mathcal{U}_2| \leq 2tP / \sin \varphi$. $\quad\square$

**Proposition 3.** *If a convex set in $\mathbb{R}^{d-1}$ contains $d - 1$ mutually orthogonal line segments of length 1, then it also contains a ball of radius $c = 1/2(d - 1)$.*

**Proof.** Denote the convex set by $D$ and the segments by $[A_i^0, A_i^1]$, $i = 1, \ldots, d - 1$. Since all points $A_i^j$ lie in $D$, each convex combination of the form $P_J = \frac{1}{d-1} \sum_{i=1}^{d-1} A_i^{J(i)}$, where $J$ denotes a map $\{1, \ldots, d - 1\} \mapsto \{0, 1\}$, also lies in $D$. The convex combination of the set of points $P_J$ is a hypercube with the size of length $1/(d - 1)$ and contains the ball of radius $1/2(d - 1)$ with the center at the hypercube's center. $\quad\square$

**Proposition 4.** *$B_t$ contains $d - 1$ mutually orthogonal line segments of length $\beta_t$, where $\beta_t / t \to \infty$ as $t \to 0$.*

**Proof.** Take a unit vector $e'$ orthogonal to $e$, and consider the 2-dimensional plane $\Pi'$ through $r_0$ parallel with $e$ and $e'$. The intersection $\Pi' \cap C =: C'$ is a 2-dimensional convex body, and $r_0$ is a regular point on its boundary; the intersection $\Pi' \cap \Pi_t = l'_t$ is a line orthogonal to $e$ at the distance $t$ from $r_0$; the intersection $\Pi' \cap B_t$ is a line segment (maybe degenerating to a point or the empty set). Equivalently, this segment is the intersection of the body $C'_t$ with the line $l'_t$. Since the point $r_0 \in \partial C'$ is regular, we conclude that the length of this segment $\beta'_t$ satisfies $\beta'_t / t \to \infty$ as $t \to 0$.

Now, choose unit vectors $e_1, \ldots, e_{d-1}$ in such a way that the set of vectors $e_1, \ldots, e_{d-1}, e$ forms an orthonormal system in $\mathbb{R}^d$. For each $i = 1, \ldots, d - 1$, draw the 2-dimensional plane $\Pi^i$ through $r_0$ parallel with $e$ and $e_i$. The intersections $\Pi^i \cap B_t$ are line segments parallel with $e_i$, and therefore, they are mutually orthogonal. The lengths of these segments

$\beta_t^i$ satisfy $\beta_t^i / t \to \infty$ as $t \to 0$. Taking $\beta_t = \min_{1 \le i \le d-1} \beta_t^i$, one comes to the statement of the proposition. $\square$

Recall that $S_t$ is the intersection of $\partial C$ with the half-space $\{r : \langle r - r_0, e \rangle \ge -t\}$ and $\Pi_t$ is the plane of the equation $\langle r - r_0, e \rangle = -t$. For $\varphi \in (0, \pi/2)$, denote by $S_{t,\varphi}$ the part of $S_t$ containing the regular points $r$ satisfying $\langle n_r, e \rangle \le \cos \varphi$. In other words, $S_{t,\varphi}$ is the set of regular points $r$ in $S_t$ such that the angle between $e$ and $n_r$ is greater than or equal to $\varphi$.

**Proposition 5.** *We have*

$$\frac{|S_{t,\varphi}|}{|B_t|} \to 0 \quad \text{as } t \to 0.$$

**Proof.** Consider a coordinate system $(x, z)$, $x = (x_1, \ldots, x_{d-1})$ such that the $x$-plane coincides with $\Pi_t$ and the $z$-axis is directed toward the vector $e$. For $t_0 > 0$ sufficiently small, the intersection of $\Pi_t$ and the interior of $C$ is nonempty for all $t \le t_0$. The angle between $-e$ and the outward normal at each regular point of $S_t$, $t \le t_0$ is greater than a positive value $\varphi_0$. That is, for any regular point $r \in S_t$, it holds $\langle n_r, e \rangle \ge -\cos \varphi_0$. Without loss of generality, one can take $\varphi < \varphi_0$, and then, for all regular points $r \in S_{t,\varphi}$, it holds $|\langle n_r, e \rangle| \le \cos \varphi$.

In the chosen coordinate system, $C_t$ is contained between the planes $z = 0$ and $z = t$. Denote by $D_t$ the image of $C_t$ under the natural projection $(x, z) \mapsto x$. The domain $D_t$ contains $B_t$ and is contained in the $(t \cot \varphi)$-neighborhood of $B_t$; hence, its $(d-2)$-dimensional volume does not exceed $P_t = |\partial B_t|_{d-2} + s_{d-2}(t \cot \varphi)^{d-2}$, where $s_{d-2} = |S^{d-2}|_{d-2}$ means the area of the $(d-2)$-dimensional unit sphere.

Applying Proposition 2 to the body $C = C_t$ and the domain $\mathcal{U} = S_{t,\varphi}$, one obtains

$$|S_{t,\varphi}| \le \frac{2t P_t}{\sin \varphi} = 2t \frac{|\partial B_t|_{d-2} + s_{d-2}(t \cot \varphi)^{d-2}}{\sin \varphi}.$$

By Propositions 3 and 4, $B_t$ contains a ball of radius $c\beta_t$, and therefore, by Proposition 1,

$$|\partial B_t|_{d-2} \le \frac{d-1}{c\beta_t} |B_t|$$

and additionally, $|B_t| \ge b_{d-1}(c\beta_t)^{d-1}$, where $b_{d-1}$ means the volume of the unit ball in $\mathbb{R}^{d-1}$. Hence,

$$\frac{|S_{t,\varphi}|}{|B_t|} \le 2t \frac{\frac{d-1}{c\beta_t}|B_t| + s_{d-2}(t \cot \varphi)^{d-2}}{\sin \varphi |B_t|} \le \frac{2(d-1)}{c \sin \varphi} \frac{t}{\beta_t} + \frac{2s_{d-2}}{c \sin \varphi \, b_{d-1}} \frac{t}{\beta_t} \to 0 \quad \text{as } t \to 0.$$

$\square$

Let us now finish the proof of Theorem 1.

Recall that $\nu_S$ is the surface area measure induced by $S$. For all $\varphi \in (0, \pi/2)$, one has

$$\nu_t = \frac{1}{|B_t|} \nu_{S_{t,\varphi}} + \frac{1}{|B_t|} \nu_{S_t \setminus S_{t,\varphi}}.$$

Proposition 5 implies that the measure $\frac{1}{|B_t|} \nu_{S_{t,\varphi}}$ converges to 0 as $t \to 0$. Indeed, for any continuous function $f$ on $S^{d-1}$,

$$\int_{S^{d-1}} f(n) \frac{1}{|B_t|} \nu_{S_{t,\varphi}}(dn) \le \max |f| \frac{|S_{t,\varphi}|}{|B_t|} \to 0 \quad \text{as } t \to 0.$$

On the other hand, the measure $\frac{1}{|B_t|} \nu_{S_t \setminus S_{t,\varphi}}$ is supported in the set in $S^{d-1}$ containing all points whose radius vector forms the angle $\le \varphi$ with $e$. It follows that each partial limit

of $\frac{1}{|B_t|} \nu_{S_t \setminus S_{t,\varphi}}$ and, therefore, each partial limit of $\nu_t$ are supported in this set. Since $\varphi > 0$ can be made arbitrary small, one concludes that each partial limit of $\nu_t$ is proportional to $\delta_e$. Finally, utilizing Equality (5) true for each partial limit $\nu_*$, one concludes that the limit of $\nu_t$ exists and is equal to $\delta_e$.

### 3. Proof of Theorem 2

In the proof, we will use the well-known fact that the surface area measure is continuous with respect to the Hausdorff topology in the space of convex bodies.

More precisely, we say that a family of convex bodies $C_t$, $t > 0$ in $\mathbb{R}^d$ converges to a convex body $C \subset \mathbb{R}^d$ as $t \to 0$ in the sense of Hausdorff and write $C_t \xrightarrow[t \to 0]{} C$, if for any $\varepsilon > 0$, there exists $t_0 > 0$ such that for all $t \leq t_0$, $C_t$ is contained in the $\varepsilon$-neighborhood of $C$ and $C$ is contained in the $\varepsilon$-neighborhood of $C_t$.

It is well known that if $C_t \xrightarrow[t \to 0]{} C$, then $\nu_{\partial C_t} \to \nu_{\partial C}$ as $t \to 0$.

Choose $\sigma > 0$ so $|\hat{B}_\sigma| = 1$, and therefore,

$$\nu_{\hat{S}_\sigma} = \nu_\star. \tag{10}$$

Let the origin coincide with the point $r_0$, that is $r_0 = \vec{0}$; then, the homothety of a set $\mathcal{A}$ with the center at $r_0$ and ratio $k$ is $k\mathcal{A}$. See Figure 4.

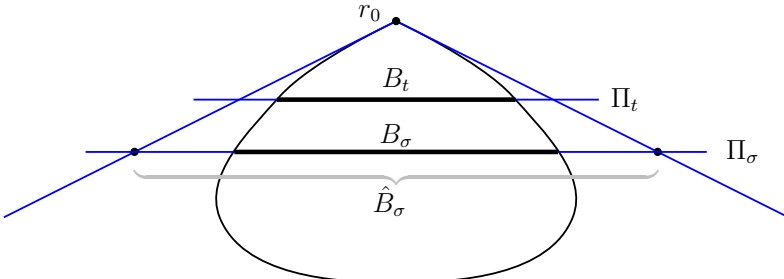

**Figure 4.** The tangent cone at $r_0$, the cutting planes $\Pi_t$ and $\Pi_\sigma$, and the sets $B_t$, $B_\sigma$, and $\hat{B}_\sigma$ in the case when the point $r_0$ is conical.

**Proposition 6.** $\frac{\sigma}{t} B_t \xrightarrow[t \to 0]{} \hat{B}_\sigma$.

**Proof.** Note that for all positive $t_1$ and $t_2$,

$$\frac{t_1}{t_2} \Pi_{t_2} = \Pi_{t_1} \quad \text{and} \quad \frac{t_1}{t_2} \hat{B}_{t_2} = \hat{B}_{t_1}.$$

Additionally, since the tangent cone $K$ contains $C$, then $\hat{B}_t$ contains $B_t$, and so,

$$\frac{\sigma}{t} B_t \subset \frac{\sigma}{t} \hat{B}_t = \hat{B}_\sigma.$$

Let now $0 < t_1 \leq t_2$. Since $\vec{0}$ and $B_{t_2}$ belong to $C$, so does their linear combination,

$$\frac{t_1}{t_2} B_{t_2} = \left(1 - \frac{t_1}{t_2}\right)\vec{0} + \frac{t_1}{t_2} B_{t_2} \subset C.$$

On the other hand, $\frac{t_1}{t_2} B_{t_2} \subset \frac{t_1}{t_2} \Pi_{t_2} = \Pi_{t_1}$. It follows that $\frac{t_1}{t_2} B_{t_2} \subset C \cap \Pi_{t_1} = B_{t_1}$. We conclude that

$$\frac{\sigma}{t_2} B_{t_2} \subset \frac{\sigma}{t_1} B_{t_1},$$

that is $\frac{\sigma}{t} B_t$, $t > 0$ form a nested family of sets contained in $\hat{B}_\sigma$.

Suppose that $\frac{\sigma}{t} B_t$ does not converge to $\hat{B}_\sigma$. This implies that the closure of the union

$$\overline{\bigcup_{t>0} \frac{\sigma}{t} B_t} =: \tilde{B}_\sigma$$

is contained in, but does not coincide with, $\hat{B}_\sigma$.

The union:

$$\bigcup_{t>0} \frac{t}{\sigma} \tilde{B}_\sigma =: \tilde{K}$$

is a cone with the vertex at $r_0$; it is contained in the tangent cone $K$, but does not coincide with it. On the other hand,

$$C = \bigcup_{t \geq 0} B_t \subset \bigcup_{t \geq 0} \frac{t}{\sigma} \tilde{B}_\sigma = \tilde{K};$$

that is $C$ is contained in the cone $\tilde{K}$, which is smaller than the tangent cone $K$. This contradiction proves our proposition. $\square$

From Proposition 6, it follows, in particular, that

$$\lim_{t \to 0} \left| \frac{\sigma}{t} B_t \right| = |\hat{B}_\sigma| = 1, \tag{11}$$

and therefore,

$$\nu_{\frac{\sigma}{t} B_t} = \left| \frac{\sigma}{t} B_t \right| \delta_{-e} \longrightarrow |\hat{B}_\sigma| \delta_{-e} = \nu_{\hat{B}_\sigma} \quad \text{as } t \to 0. \tag{12}$$

Denote

$$\Sigma_\sigma^t := \mathrm{conv}\left( \frac{\sigma}{t} B_t \cup r_0 \right).$$

Since the convex body $\frac{\sigma}{t} C_t$ contains both $r_0$ and $\frac{\sigma}{t} B_t$, we have $\Sigma_\sigma^t \subset \frac{\sigma}{t} C_t$.

Recall that $K_\sigma$ is the part of the tangent cone cut off by the plane $\Pi_\sigma$. We have $K_\sigma = \mathrm{conv}(\hat{B}_\sigma \cup r_0)$. Since by Proposition 6, $\frac{\sigma}{t} B_t \xrightarrow[t \to 0]{} \hat{B}_\sigma$, we conclude that $\mathrm{conv}\left( \frac{\sigma}{t} B_t \cup r_0 \right) \xrightarrow[t \to 0]{} \mathrm{conv}(\hat{B}_\sigma \cup r_0)$, that is

$$\Sigma_\sigma^t \xrightarrow[t \to 0]{} K_\sigma.$$

Using this relation and the double inclusion:

$$\Sigma_\sigma^t \subset \frac{\sigma}{t} C_t \subset K_\sigma,$$

one concludes that $\frac{\sigma}{t} C_t$ converges to $K_\sigma$ in the sense of Hausdorff, and therefore,

$$\nu_{\frac{\sigma}{t} \partial C_t} \to \nu_{\partial K_\sigma} \quad \text{as} \quad t \to 0.$$

Using that $\frac{\sigma}{t} \partial C_t = \frac{\sigma}{t} S_t \cup \frac{\sigma}{t} B_t$ and $\partial K_\sigma = \hat{S}_\sigma \cup \hat{B}_\sigma$ and using (12), one obtains

$$\nu_{\frac{\sigma}{t} S_t} \to \nu_{\hat{S}_\sigma} \quad \text{as} \quad t \to 0,$$

and taking account of (11), one obtains

$$\lim_{t \to 0} \nu_t = \lim_{t \to 0} \frac{1}{|B_t|} \nu_{S_t} = \frac{1}{\lim_{t \to 0} \left| \frac{\sigma}{t} B_t \right|} \lim_{t \to 0} \nu_{\frac{\sigma}{t} S_t} = \nu_{\hat{S}_\sigma} = \nu_\star.$$

Theorem 2 is proven.

**Funding:** This research received no external funding.

**Institutional Review Board Statement:** Not applicable.

**Informed Consent Statement:** Not applicable.

**Data Availability Statement:** Not applicable.

**Acknowledgments:** This work was supported by the Center for Research and Development in Mathematics and Applications (CIDMA) through the Portuguese Foundation for Science and Technology (FCT), within Projects UIDB/04106/2020 and UIDP/04106/2020.

**Conflicts of Interest:** The author declares no conflict of interest.

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
