# Peer review of "Local Structure of Convex Surfaces near Regular and Conical Points"

_axioms, doi:10.3390/axioms11080356_

Round 1

Reviewer 1 Report

This paper deals with a question related to the limiting behavior of certain surface measure on the boundary of a convex, compact set C. Given a point in the boundary, it is studied with certain detail the afore mentioned surface measure of C, when a plane of support to the surface at such a point approaches the point. It is considered two type of points in this boundary: regular points, and points where there is a kind of conical singularity (well explained in the bulk of the paper).

An interesting point in this work is the analysis made, by obtaining some adequate estimates for the intersection of C with the approaching plane (in the case of regular points) and properties of Hausdorff topology in the space of convex bodies (for the case of conical points).

This reviewer considers that this work may be of some interest of a variety of researchers by the techniques adopted, and the connection with extensions of classical variational problems.

My recommendation is that the manuscript should be published.

Author Response

Dear Reviewer,

Many thanks for reviewing my manuscript.

Best wishes,

Alexander Plakhov

Reviewer 2 Report

This paper presents an interesting study on the Local structure of convex surfaces near regular and conical points. The paper fits well with the submitting journal. The paper is well organized and I suggest accept it in its present form.

Author Response

(The authors gave the same response as above.)

Reviewer 3 Report

Dear Editor,

I wish to recommend that this paper will be accepted for publication.

The paper studies a generalized version of Newton's problem of least resistance for a convex body moving in a very rarified medium. In particular, the paper studies the surface area measure induced by the local vicinity of a point located on the surface of a convex body, both when the point is regular and also when it is singular. 

The paper is well written: the introduction provides the motivation for studying the problem and presents the tools needed for studying it, together with a summary of previous works. The results are precisely stated and clearly presented and proved through a well-organized set of propositions. 

Author Response

(The authors gave the same response as above.)
